# Anthracene and Pyrene Biodegradation Performance of Marine Sponge Symbiont Bacteria Consortium

**DOI:** 10.3390/molecules26226851

**Published:** 2021-11-13

**Authors:** Ismail Marzuki, Ruzkiah Asaf, Mudian Paena, Admi Athirah, Khairun Nisaa, Rasheed Ahmad, Mudyawati Kamaruddin

**Affiliations:** 1Department of Chemical Engineering, Fajar University, Makassar 90231, Indonesia; 2Research Center for Brackish Aquaculture Fisheries and Extension Fisheries, Maros 90512, Indonesia; qiaasaf@gmail.com (R.A.); mudianpaena@yahoo.co.id (M.P.); m.athirah@gmail.com (A.A.); 3Fishery Faculty, Cokroaminoto University of Makassar, Makassar 90245, Indonesia; nisauicha27@gmail.com; 4Departement of Chemistry, Airlangga University, Surabaya 60115, Indonesia; rasheed.ahmad@fst.unair.ac.id; 5Postgraduate Program, Department of Medical Laboratory Science, Muhammadiyah Semarang University, Semarang 50273, Indonesia; mudyawati@unimus.ac.id

**Keywords:** performance, biodegradation, bacterial consortium, marine sponge, PAHs

## Abstract

Every petroleum-processing plant produces sewage sludge containing several types of polycyclic aromatic hydrocarbons (PAHs). The degradation of PAHs via physical, biological, and chemical methods is not yet efficient. Among biological methods, the use of marine sponge symbiont bacteria is considered an alternative and promising approach in the degradation of and reduction in PAHs. This study aimed to explore the potential performance of a consortium of sponge symbiont bacteria in degrading anthracene and pyrene. Three bacterial species (*Bacillus pumilus* strain GLB197, *Pseudomonas stutzeri* strain SLG510A3-8, and *Acinetobacter calcoaceticus* strain SLCDA 976) were mixed to form the consortium. The interaction between the bacterial consortium suspension and PAH components was measured at 5 day intervals for 25 days. The biodegradation performance of bacteria on PAH samples was determined on the basis of five biodegradation parameters. The analysis results showed a decrease in the concentration of anthracene (21.89%) and pyrene (7.71%), equivalent to a ratio of 3:1, followed by a decrease in the abundance of anthracene (60.30%) and pyrene (27.52%), equivalent to a ratio of 2:1. The level of pyrene degradation was lower than that of the anthracene due to fact that pyrene is more toxic and has a more stable molecular structure, which hinders its metabolism by bacterial cells. The products from the biodegradation of the two PAHs are alcohols, aldehydes, carboxylic acids, and a small proportion of aromatic hydrocarbon components.

## 1. Introduction

Anthracene and pyrene are polycyclic aromatic hydrocarbons (PAHs). Their molecular structure is stable due to the ring’s resonance ability of pi (π) bonds. These two types of PAHs were first isolated from coal tar and as volatile compounds formed from the incomplete combustion of organic compounds in protein-rich foods [1]. The primary sources of anthracene and pyrene are coal tar, natural gases, and petroleum. These PAHs are also present in plants (e.g., tobacco tar) and animals (e.g., deer and termites) [2]. Anthracene is moderately toxic and carcinogenic, whereas pyrene is highly toxic, carcinogenic, and genotoxic or mutagenic [3,4,5]. Anthracene is widely used as an antiseptic and insecticide, whereas pyrene is used for increasing the octane number of fossil fuels [6,7]. The current use of anthracene and pyrene is quite extensive; hence, there is a risk of them being pollutants in the air, soil, and water [8,9]. As one of the characteristics of PAHs, anthracene and pyrene are difficult to decompose. They accumulate in the environment, including in living organisms. Supposedly, these PAHs have been released in the environment for a long time, which, in combination with the effects of climate factors such as precipitation and temperature, can disrupt the ecological balance [10]. Additionally, these two PAHs can be dissolved in water bodies, eventually ending up in the sea [11].

The sea is a giant container that can accommodate a variety of materials. There is a wide range of biota interacting in the sea, including the reversible relationship between all abiotic components and biotic components, i.e., those already available in the sea and new materials that enter the marine environment [12,13]. This condition results in new adaptations in the sea, resulting in changes in the life patterns of marine ecosystems [14,15]. If the discharge of PAHs into the sea is not properly controlled, fish and other biota will accumulate these carcinogenic components [16,17]. These fish are then caught by fishermen and eventually consumed by humans [18]. This process eventually forms a cycle, and the flow of the food chain, in the long run, decreases both environmental quality and human health [19]. 

The formation of interactive patterns in the sea between abiotic and biotic components as the result of adaptation to the above conditions causes many organisms to experience pressure, thereby regulating their lifecycle. However, some organisms thrive due to these new patterns [20,21]. One example is the sea sponge because it has a strong ability to adapt to changes in habitat [22]. There are four forms of adaptation by marine sponges. *First*, sponges have the ability to form a mutual symbiosis with bacteria such that they can survive in habitats contaminated with waste such as PAHs and heavy metals [23]. *Second,* the nutritional pattern of sponges is a filter feeder with a body structure that has oscula, enabling it to obtain and absorb food that is needed, whereas food that is not needed is disposed of by spraying it out [24]. *Third,* sponges can produce mucus substances that act as enzymes, which can be spread on the surface of their bodies to camouflage them against predatory threats [25,26]. Several earlier investigations discovered that any bacteria with the potential to biodegrade PAHs were isolated from sponges having mucus on their body surface. This mucus is thought to be utilized by sponges to defend themselves against predators in their environments, including the presence of poisonous PAHs and heavy metals [11,26]. *Fourth*, sponges are able to absorb carbon components and convert them into energy for activities, including being able to absorb heavy metals [27,28,29,30]. Some scientific data have suggested this adaptability of sponges to be due to their old civilization. The DNA structure of sponges responds easily to habitat changes [31].

In addition to having the ability to degrade carbon components, especially PAHs, sponges also have bioindicator and biomonitoring functions, as well as the ability to bio-adsorb heavy metals. This is due to their ability to produce enzymes (metabolic components) that can absorb and neutralize the toxic nature of the waste [32,33,34]. Sponges are also able biodegrade aliphatic and aromatic hydrocarbon components, presumably because of the role of symbiotic bacteria or because of the metabolic components produced [35,36]. The biodegradation of PAHs and bio-adsorption of heavy metals have led to the marine sponge being identified with significant benefits in reducing pollutants and maintaining the quality of the marine environment [37,38]. 

The facts mentioned above prompted us to carry out a series of studies to explore and map the abilities of sponges and their symbiont bacteria to reduce PAHs and heavy-metal pollutants in the marine environment. Thus, data on the collection and even formulation of marine sponge symbiont bacteria can be compiled with their biodegradation and bio-adsorption abilities to form a crystalline bacterial consortium that can be quickly mobilized to the polluted sites [1,11,39]. Characterization of sponges and symbiotic bacteria, including an analysis of the biodegradation potential of PAHs, is essential [40]. The application and increase in the biodegradation rate of symbiont bacteria toward hydrocarbon pollutants, especially PAHs, is seen as a necessity [41,42].

Several species of sponge symbiont bacteria can also carry out the bio-adsorption of several types of heavy metals, namely, mercury (Hg), chromium (Cr), arsenic (As), lead (Pb), cadmium (Cd), copper (Cu), nickel (Ni), zinc (Zn), and cobalt (Co) [43,44,45,46]. The development of the function and performance in the biodegradation of PAHs and bio-adsorption of heavy metals by sponge symbiont bacteria, in the form of a consortium of potential bacteria, can be used to effectively manage polluted environments [47]. A previous study also looked at several types of sponge symbiotic bacteria with the capacity to biodegrade PAH components and bio-adsorb heavy metals [24]. Following a series of effective studies, future environmental management operations to remove PAHs and heavy metals can be carried out using bacteria that have been shown to breakdown PAH pollutants and absorb heavy metals. The maritime environment is severely polluted with both sorts of pollutants (PAHs and heavy metals); therefore, materials that can address this are required [48]. The role of sponge symbiotic biomass and bacteria in improving the quality of the marine environment is of great interest. The biodegradation performance of PAHs can be improved by utilizing a bacterial consortium. Possible practical applications allow preserving the environment while maintaining the growth and development of the marine sponge population. The cultivation of marine sponges can be achieved using the transplantation method [49,50,51]. 

The major goal of this study was to evaluate a consortium of bacteria for its biodegradation of anthracene and pyrene by examining the level of biodegradation products, in addition to biodegradation metrics such as turbidity, gas generation, changes in pH, and fermentation odor.

## 2. Results

This study attempted to observe the ability of a consortium formulated from marine sponge symbiont bacteria to decompose anthracene and pyrene. This included the preparation of the marine sponge symbiont bacteria consortium through a characterization of the sampling point, morphology analysis, bacterial isolation from marine sponges, phenotypic and genotypic analyses of the isolated bacterial, and selection of potential PHAs for biodegradation, followed by an evaluation of the biodegradation processes of the bacterial consortium as a function of various parameters including the reduction in abundance and concentration of the PHAs, and a measurement of the biodegradation products or functional groups.

### 2.1. Characteristics of Seawater at Sampling Location

The seawater at the point of sponge sample acquisition was characterized to establish any potential correlation between the habitat of the sponge (e.g., the presence of hazardous contaminants and potential symbiotic bacteria for their biodegradation). The sampling points were the waters around Kodingareng Keke Island, in the administrative area of Makassar Metro City, South Sulawesi. The island is one of the Maritime Tourism Areas developed by the Makassar City government and is included in the Spermonde Archipelago Cluster [11]. Table 1 shows the characteristics of the sampling locations. The data obtained in the table shows the general condition of coastal waters.

### 2.2. Marine Sponge Morphology

The morphology of marine sponges, as the source of bacterial isolates for the biodegradation of PAHs, was evaluated in terms of the sponge body and cell structure. The bacterial symbiotic model exhibited potential for biodegradation of PAHs. Sponges are marine organisms with many functions, but they are vulnerable to threats resulting from predators or fishing activity disturbances due to the slow growth rate and development of sponges. It is necessary to understand whether there is a relationship between the body structure of marine sponges and their associated bacteria, as well as between their habitats and the presence of slimy body surfaces with the ability to decompose PAH-type hydrocarbon components using the symbiotic bacteria isolated from the sponges. Marine sponge samples coded Sp1, Sp2, and Sp3, according to the results of morphological analysis, were *Niphates* sp. (Figure 1), *Hyrtios erectus* (Figure 2), and *Clathria (Thalysias) reinwardtii* (Figure 3), respectively. Interestingly, all three marine sponges presented a slimy surface [11,39].

In general, the features of these three species of sponges were similar, while their body surfaces were covered with mucus and supplied various types of bacteria. Other studies conducted using bacteria obtained from marine sponges whose bodies are not coated with mucus substances in their natural habitat resulted in these bacteria being unable to breakdown PAHs. This validates the idea that bacteria isolated from marine sponges with mucus-covered body surfaces can be employed for the biodegradation of PAHs.

### 2.3. Isolation and Phenotypic Characteristics of Marine Sponge Symbiont Bacteria 

Potential bacteria for the biodegradation of PAHs can be isolated from marine sponges. A phenotypic citrate test was preliminarily performed to confirm whether the chosen bacteria had the potential to biodegrade PAHs. A positive indicates that the bacteria can likely convert carbon as an energy source and are, thus, potentially able to biodegrade PAHs.

Phenotypic characterization of three bacterial isolates showed that the bacteria formed spores, were Gram-positive, and generally reacted positively to seven types of biochemical reagents These characteristics were considered to meet the primary requirements to carry out the biodegradation process of PAH components [9,31]. Several conditions are needed so that bacteria can carry out the biodegradation of hydrocarbon components, e.g., having enzymes capable of hydrolyzing amino acids as indicated by the indole test, ability to carry out the fermentation reaction as shown by the TSIA test, ability to reduce nitrate to nitrile as shown by the nitrate test, and ability to break bonds as shown by the urease test. The urease test indicated the presence of carbon, while the positive citrate test indicated that bacteria can convert citrate to pyruvate, which enters the organism’s metabolic cycle to produce energy. On the basis of these results, we can predict that these bacteria can also biodegrade PAHs. Although not all bacteria examined exhibited positive results for all tests, other criteria suggested the potential utilization of these bacteria in the biodegradation of PAH components. However, no particular phenotypic feature showing that sponge symbiotic bacteria may breakdown PAHs was identified. Selected bacteria exhibited more than two phenotypic characteristics for the biodegradation of PAHs from seven phenotypic tests (Table 2). The phenotypic analysis also revealed that the bacterial isolate coded Sp1, isolated from the sponge *Niphates* sp., belongs to the *Bacillus* group, Sp2, isolated from the sponge *Hyrtios erectus*, belongs to the *Pseudomonas* group, and Sp3, isolated from the sponge *Clathria* (*Thalysias*) *reinwardtii*, belongs to the *Acinetobacter* group [22,24,29].

### 2.4. Genotypic Analysis of Marine Sponge Symbiont Bacteria 

It is necessary to know the bacterial genotype of marine sponge symbionts to ascertain the types and strains of bacteria used to degrade PAH components. According to the data on the grouping of marine sponge symbiont bacteria (Table 2), the three isolates were assigned a new code as a function of the suspected bacterial class, i.e., Sp1-Bc, Sp2-Ps, and Sp3-Ac. The results of genotypic characterization using the 16S rRNA gene sequence indicated that bacterial isolate Sp1-Bc had a 96% similarity level with *Bacillus pumilus* strain GLB197, the Sp2-Ps isolate had a 91% similarity level with *Pseudomonas stutzeri* strain SLG510A3-8, and the Sp3-Ac isolate had a 99% similarity level with *Acinetobacter calcoaceticus* strain SLCDA976 (Table 3).

### 2.5. Biodegradation of PAH Components

#### 2.5.1. Evaluating PAH Biodegradation Performance of the Marine Sponge Symbiont Bacterial Consortium

The physical interaction of the bacterial consortium suspension with the PAHs changed during the biodegradation process. Several general parameters can be observed as indicators of the ongoing biodegradation process, including turbidity, fermentation odor, pH changes, and the formation of gas bubbles (Table 4). These parameters are characteristic of fermentation by bacteria or the actions of enzymes on a substrate containing protein or glucose, including materials containing hydrocarbon components. Biodegradation follows the pattern of enzymatic reactions. The enzymatic components are thought to be produced by bacteria in response to their habitat conditions contaminated with hydrocarbon components. Mucus with enzyme characteristics is produced for bacteria to survive in PAH-contaminated media. Therefore, physical changes in the interaction media denote the performance of enzymatic reactions via the appearance of fermentation odor and the abundance of gas bubbles [29,30,31].

The relative turbidity of the media increased with increasing interaction time, indicating that there was an increase in the size and number of bacterial cells. The increase in temperature, the relative increase in acidity of the interactive media, the emergence of gas bubbles, and the smell of fermentation are strong indications of a biodegradation process occurring in the media. These parameters are characteristic of the fermentation process of enzymes produced by marine sponge symbiont bacteria in response to media contaminated with anthracene and pyrene components. These results indicate that the consortium of marine sponge symbiont bacteria can degrade anthracene and pyrene components [37,42].

#### 2.5.2. Analysis of the Biodegradation Performance of Anthracene and Pyrene PAHs by Marine Sponge Symbiont Consortium

Anthracene and pyrene act as substrates degraded by the bacterial consortium. In theory, the number of peaks detected by GC–MS should only be two, corresponding to the presence of anthracene and pyrene in the mixture. However, this was not the case, as more peaks were recorded. A longer interaction period led to more new peaks appearing, indicating that those new peaks are products resulting from the biodegradation of PAHs [11,42,52]. The new peaks were attributed to alcohols, aldehydes, carboxylic acids, some ketones, and aliphatic hydrocarbon compounds. However, this cannot be estimated with absolute certainty because the quality of some GC–MS components was below 90% (Table 5).

The peaks of anthracene and pyrene decreased with the increase in interaction time. This result indicates that the medium’s abundance of anthracene and pyrene components was reduced due to degradation. Under the same conditions, the number of new components tended to increase, and the total concentration of biodegradation products also increased.

The abundance of anthracene and pyrene components seemed to decrease with increasing interaction time. The decrease in abundance of the component correlated with the decrease in component peak. Subsequently, new peaks appeared, indicating new products of the biodegraded compound. The number of new peaks formed tended to increase with interaction time. These new peaks had different retention times and peak heights, indicating different types and abundances [1,4,53]. Each new component had different characteristics, i.e., functional groups, but they were all organic compounds. Anthracene and pyrene components are shown in red in Figure 4A–E, while degradation products components are shown in green. Analysis of the degraded components on the spectra (Figure 4A–E) showed differences in the number and type of component. Accordingly, it can be stated that (1) the number of components resulting from biodegradation tended to increase with increasing interaction time, (2) the types of biodegradation products were different, and (3) all biodegradation products were organic compounds with dominant characteristics in the form of hydroxyl and carbonyl functional groups.

The maximum biodegradation process was characterized by a decrease in peak height or a decrease in the abundance of PAHs that occurred following a contact time of 15–20 days (Figure 4C,D), after which the biodegradation process was relatively stagnant (Figure 4E).

#### 2.5.3. Functional Groups of Biodegradation Products 

The FTIR spectrum (Figure 5) presents the components of biodegradation products in the form of organic compounds. Each compound is characterized by functional groups identified according to the range of wavenumbers. Overall, it can be stated that a longer interaction time led to an increase in the number of components and the types of components (Figure 5A–E), dominated by alcohols, aldehydes, ketones, carboxylic acids, and some aromatic components [5,8,11]

### 2.6. Comparison of the Biodegradation of Anthracene and Pyrene by the Marine Sponge Symbiont Consortium Bacteria

The performance and level of biodegradation of anthracene by the marine sponge symbiotic bacterial consortium were different from those of pyrene. With an interaction time of 25 days (Figure 6A), the decrease in the abundance of anthracene reached 4.132 × 10^6^ units, equal to 60.30%, while that of pyrene reached 7.902 × 10^6^ units, equal to 27.52%, i.e., a ratio of 1:2. The decrease in concentrations of anthracene and pyrene (Figure 6B) exhibited a ratio of 1:3, i.e., the percentage decrease in the concentration of anthracene was 21.89%, while that of pyrene was only 7.71%. The total concentration of PAHs (Ant. + Pyr.) degraded by the bacterial consortium was 17.23% (Figure 6C), while the total concentration of biodegradation products formed was 17.67%. These results indicate that the consortium of marine sponge symbiotic bacteria more easily degraded anthracene than pyrene [2,11,54].

## 3. Discussion

The seawater conditions at the sponge sampling point were of good quality (Table 1) according to salinity, pH, electrical conductivity, and total dissolved solids (TDS). However, this does not suggest that the seawater is free from hydrocarbon contamination. The results of the morphological analysis showed that all three types of marine sponges as sources of bacterial isolates had a slimy body surface (Figure 1, Figure 2 and Figure 3). The mucus on the surface of the sponge body is anticipated to be a substance produced by symbiotic bacteria for self-protection and adaptation to the dynamics in the sponge’s growth habitat [5,15,55]. The three types of consortium bacteria (Table 2) reacted positively with several phenotypic test reagents, indicating their specific ability to biodegrade PAHs, such as pyrene and anthracene. The phenotypic information was supplemented by genotypic analysis in the form of strains and bacterial species. As a result, the features of the bacterial consortia involved in the biodegradation of anthracene and pyrene components were more evident. Genotypic analysis was carried out to ascertain the characteristics of the three types of isolates used for the biodegradation of PAHs (Table 3).

One of the novelties of this research is that it provides an understanding of potential bacteria from marine sponges that can biodegrade hydrocarbon components, especially PAHs. It can be used as guideline for compiling data on bacteria that biodegrade PAHs [11,37,39].

The observations of turbidity, temperature, pH, gas bubbles, and the fermentation smell suggested that the consortium of marine sponge symbiont bacteria was able to degrade PAH components (anthracene and pyrene) (Table 4) [56]. Further evidence of the degradation of anthracene and pyrene by the consortium of sponge symbiont bacteria was the decrease in their abundance after an interaction lasting several days (Table 5), as a function of the peak height (Figure 4A–E), changes in the percentage of degraded components (Table 5), formation of new peaks corresponding to biodegradation products (Figure 4), and percentage of biodegradation products (Table 5). The new peaks formed were attributed to simple organic compounds in the form of alcohols, aldehydes, ketones, carboxylic acids, and aromatic components (Table 4; Figure 5) [2,40,57].

Anthracene was more easily degraded by the consortium of marine sponge bacterial symbionts than pyrene (Figure 6). Theoretically, this can be explained as follows: (1) the toxicity level of pyrene is higher than anthracene; thus, bacterial cells do not survive as long during the biodegradation of pyrene compared with anthracene; (2) the structure of the pyrene molecule is more stable than that of anthracene, thereby hindering its metabolism by bacteria; (3) pyrene has more aromatic rings and forms a compact structure, thus necessitating a longer time for degradation; (4) acidic compounds are formed as degradation products, thereby limiting the biodegradation process. In these conditions, bacterial cells can experience sudden mass death or struggle to further replicate [7,16,37,58].

The combination of all analytical parameters revealed a drop in anthracene concentration of 21.89%, while the decrease in pyrene concentration was 7.71%, i.e., an equivalent ratio of 3:1. These findings were supplemented by a reduction in the abundance of anthracene (which fell to 60.30%) and pyrene (which fell to 27.52%). All biodegradation products (alcohols, aldehydes, carboxylic acids, ketones, and a small number of aromatic components) were formed at a rate of 17.67%. Because the consortium bacteria’s level of biodegradation of anthracene was higher than that of pyrene, it was hypothesized that anthracene is less toxic than pyrene. Furthermore, pyrene’s molecular structure is more stable than anthracene’s, limiting its metabolism by bacterial cells. The application of a bacterial consortium was not successful in raising the degree of PAH biodegradation, likely owing to competition between bacterial cells of various species, such that the capacity of PAHs to adapt to their hazardous environment was not observed.

## 4. Materials and Methods

### 4.1. Materials

The materials used were three types of marine sponges coded Sp1 (*Niphates* sp.), Sp2 (*Hyrtios erectus*), and Sp3 (*Clathria (Thalysias) reinwardtii*) (Figure 1, Figure 2 and Figure 3), which were collected by the researchers [11,29,42]. Three different bacterial stocks were utilized, and their genotypes were determined using the PCR technique. Total DNA from sponge symbionts was extracted and amplified using a universal primer of the 16S rRNA gene with the forward sequence 5′–CCAGCAGCCGCGGTAATACG–3′ attached to nucleotide bases 518–537 and the reverse sequence 5′–ATCGG(C/T)TACCTTGTTACGACTTC–3′ attached to nucleotide bases 1513–1491 [11,28]. Marine sponge symbiont bacteria Sp1-Bc, Sp2-Ps, and Sp3-Ac (Table 3), methanol pa, *n*-hexane for GC, anthracene CAS number 000120-12-7 and pyrene CAS number 000129-00-0 (Supelco, Bellefonte, PA, USA), Na_2_SO_4_ pa, aquabides, physiological 0.9% NaCl (commercially obtained, Supelco), materials for sponge morphology analysis, materials for standard biochemical tests following the guidelines in [59], and materials for genotypic analysis of bacterial isolates were also obtained. Bacterial isolates of marine sponge symbionts were sourced from [11,29,32,37].

### 4.2. Sampling Point Characterization

Parameters observed and measured for the seawater conditions at the sponge sampling point included coordinates, salinity, pH, TDS, EC, distance from the sampling point to the nearest shoreline, and sampling depth from sea level (Table 1). All data displayed were not previously published. The determination of sampling points and sampling techniques were guided by fishery and marine experts belonging to nongovernmental organizations concerned with marine life.

### 4.3. Marine Sponge Morphology

Morphology, cell structure, and types of sea sponge samples in the study were analyzed at the Microbiology Laboratory of Sebelas Maret University, Surakarta, Central Java. Sponge types and morphological data were sourced from [11,29,32,37,42].

### 4.4. Isolation and Phenotypic Analysis of Marine Sponge Symbiont Bacteria

Isolation of bacteria was done using the swab method. Purification was done using the direct plating method. Morphological analysis was done by direct observation. Gram staining, biochemical analysis, and identification of the isolates of marine sponge symbiont bacteria followed [59]. The data contained in Table 2, refer to previous research [29,32,37].

### 4.5. Genotypic Analysis of the Bacterial Isolate

Characteristics of marine sponge symbiont bacteria were determined through genotypic analysis using the PCR method (Table 3). The data displayed were sourced from [11,29,32,37,42].

### 4.6. Biodegradation Interactions and Evaluation of the Processes

#### 4.6.1. Interaction of Biodegradation Components

Selected bacterial isolates (Sp1-Bc, Sp2-Ps, and Sp3-Ac) from [11,29,37] were cultured. A volume of 10 mL of each bacterial suspension was put in an Erlenmeyer flask, diluted to 100 mL, homogenized, and adapted for 1 × 24 h in an incubator. To mix the bacterial suspension (consortium), 5 mL was serially pipetted in a row of 15 test tubes, before incubating for 1 × 2 h in an incubator. A total of 2 mL of mixed PAHs (Ant + Pyr) were put in each test tube previously been filled with bacterial suspension. Bacterial suspensions and PAHs were allowed to interact on a shaker incubator at 200 rpm. The interaction time was 25 days, with observations and biodegradation parameters recorded every 5 days [4,11,58].

#### 4.6.2. Parameters of Biodegradation of PAHs by Bacteria 

Measurement of biodegradation parameters for the bacterial suspensions interacting with PAHs was carried out every 5 days using appropriate instruments, in addition to direct observations of the abundance of gas bubbles formed and the odor of fermentation (Table 4) [11,12].

#### 4.6.3. Biodegradation Performance of PAHs by the Bacterial Consortium 

Measurement of the abundance and components of biodegradation products was carried out every 5 days by extracting components of anthracene and pyrene that were not degraded, as well as components of biodegradation products, using *n*-hexane. The *n*-hexane extract was dehydrated using Na_2_SO_4_ for GC–MS (Table 5, Figure 4A–E) [1,37].

#### 4.6.4. Detection of Functional Groups for Identification of Biodegradation Products of PAHs by Bacteria Consortium 

An aqueous-free *n*-hexane extract was partially used for the determination of functional groups of each biodegradation product using an FTIR instrument (Figure 5A–E). The FTIR (Fourier-transform infrared) spectra revealed the presence of organic components, such as alcohols, aldehydes, and carboxylic acids. 

### 4.7. Comparison of Biodegradation Rates of Anthracene and Pyrene 

The biodegradation level of anthracene and pyrene by the bacterial consortium was determined through an analysis of the decrease in peak height of the degraded components as a function of the interaction time.
(1)$$Abund.\;of\;degr.\;comp.\;\left(\%\right)=\frac{\left(peak\;height\right)no-\left(peak\;height\right)nt}{\left(peak\;height\right)no}\times 100\;\%$$
where no and nt represent the initial and final peak heights of the biodegradation process [7,14], respectively. The decrease in the concentration of the substrate component (anthracene or pyrene) that underwent biodegradation was determined as follows:
(2)$$Conc.of\;degr.\;comp.X\;\left(\%\right)=\frac{\;\left(initial\;conc.\;of\;comp.\;X\right)-\left(final\;conc.\;of\;comp.\;X\right)}{\left(Total\;conc.\;of\;comp.\right)}\times 100\;\%$$where *X* is the component of anthracene or pyrene PAHs that underwent degradation following an interaction time of 25 days [31]. The concentrations of biodegradation products were determined as follows:
(3)$$Biodeg.\;product\;comparison\;\left(\%\right)=\frac{Total\;comp.\;of\;biodegr.\;products}{Total\;conc.\;of\;comp.}\times 100\;\%$$

Equation (1) was used to calculate the degradation rate of anthracene and pyrene by a consortium of bacteria as a function of the recorded peak height data. The data was presented in Figure 4A–E and Figure 6A and Table 5. Equation (2) was used to determine the concentration of the degraded anthracene and pyrene components. The data was presented in Figure 6B. Equation (3) was used to determine the total concentration of biodegradation products. The data was presented in Figure 6C. Each result was compared to obtain the biodegradation ratio.

## 5. Conclusions

According to the results and their analysis, several conclusions can be drawn. The bacterial consortium succeeded in degrading 21.89% of anthracene and 7.71% of pyrene. The total concentration of PAHs reached 17.67%. Several types of biodegradation products were found in this study, including alcohols, aldehydes, carboxylic acids, ketones, and a small number of aromatic components. The use of a bacterial consortium was less effective in increasing the level of biodegradation of the tested PAHs, compared with the use of a single bacterium. This was presumably due to competition between bacterial cells in the media. Acidic biodegradation products limited the ability of bacterial cells to continue the biodegradation process. The research indicates that sponge symbiont bacteria whose body surface is covered with mucus have the potential to biodegrade PAHs.

## 6. Patents

There are three patents described in this manuscript in the public testing process stage, as recorded by the Ministry of Law and Human Rights of the Republic of Indonesia: (1) performance of microsymbionts from marine sponge culture as a biodegradator of polycyclic aromatic hydrocarbons (PAH), code registration: P15202008653; (2) tracing method for bacterial isolates isolated from marine sponges as biodegradator material for polycyclic aromatic hydrocarbons (PAH), registration code: S15201907661; (3) new metal-lo-clastic bacterial species cultured from marine sponges as biomaterials for adsorption of several kinds of heavy metals, registration code: P15202008653. 

## Figures and Tables

**Figure 1 molecules-26-06851-f001:**
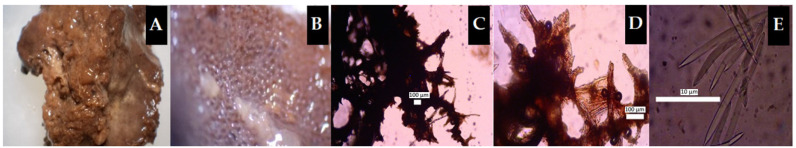
The morphology of marine sponge *Niphates* sp. (Sp1): (**A**) consistency; (**B**) surface; (**C**) skeleton; (**D**) skeletal tract; (**E**) spicules (40×).

**Figure 2 molecules-26-06851-f002:**
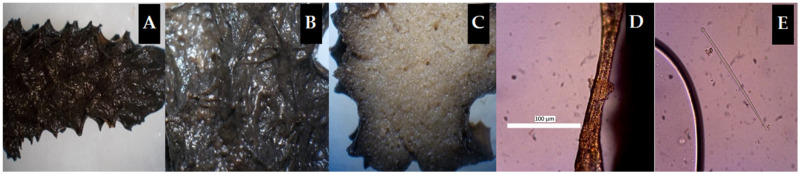
The morphology of marine sponge *Hyrtios erectus* (Sp2): (**A**) consistency; (**B**) surface; (**C**) choanosomes; (**D**) skeleton; (**E**) spicules (10×).

**Figure 3 molecules-26-06851-f003:**
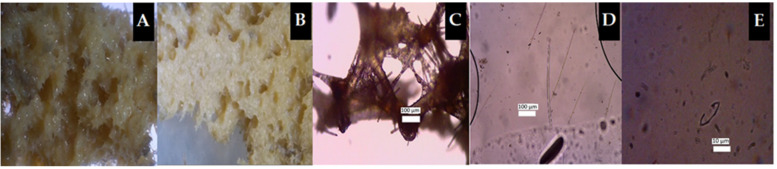
Morphology of the marine sponge *Clathria (Thalysias) reinwardtii* (Sp3): (**A**) consistency; (**B**) projection; (**C**) choanosome; (**D**) megasclere; (**E**) microsclera in the form of chelae (40×).

**Figure 4 molecules-26-06851-f004:**
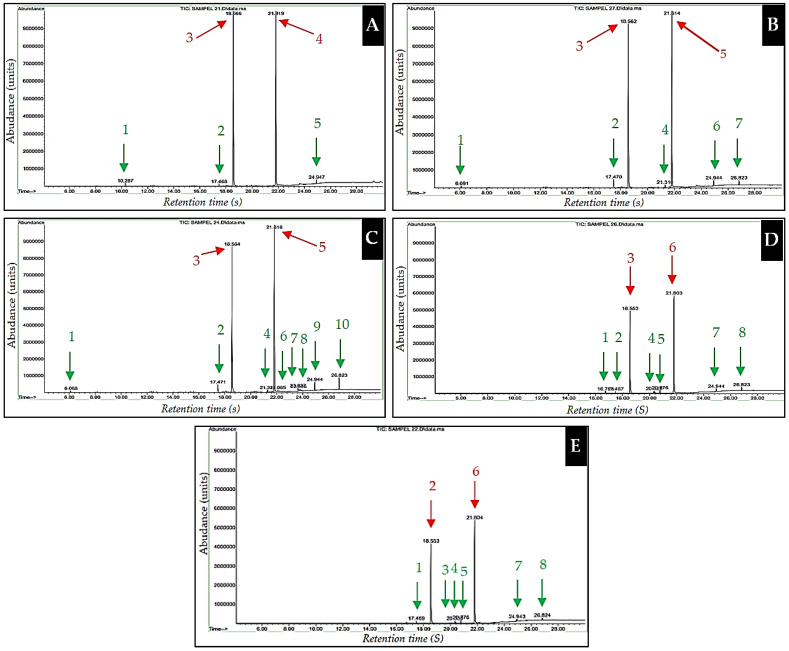
GC–MS spectrum showing the biodegradation performance of the marine sponge consortium bacterial suspension in terms of the abundance of anthracene and pyrene components and new peaks related to biodegradation products as a function of interaction time: peak abundance after (**A**) 5 days of interaction, (**B**) 10 days of interaction, (**C**) 15 days of interaction, (**D**) 20 days of interaction, and (**E**) 25 days of interaction.

**Figure 5 molecules-26-06851-f005:**
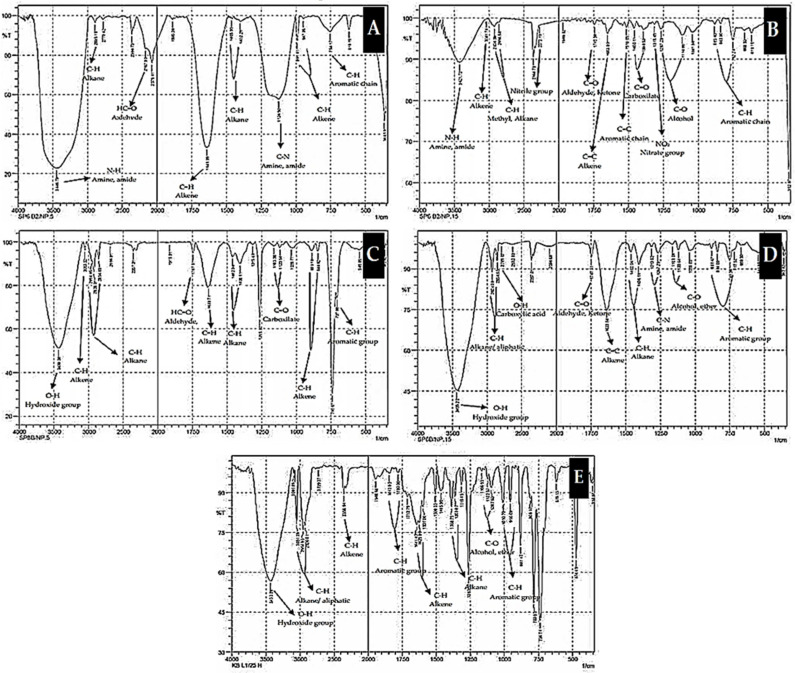
FTIR spectra displaying the functional groups of biodegradation products as a function of the interaction time: (**A**) 5 days of interaction, (**B**) 10 days of interaction, (**C**) 15 days of interaction, (**D**) 20 days of interaction, and (**E**) 25 days of interaction.

**Figure 6 molecules-26-06851-f006:**
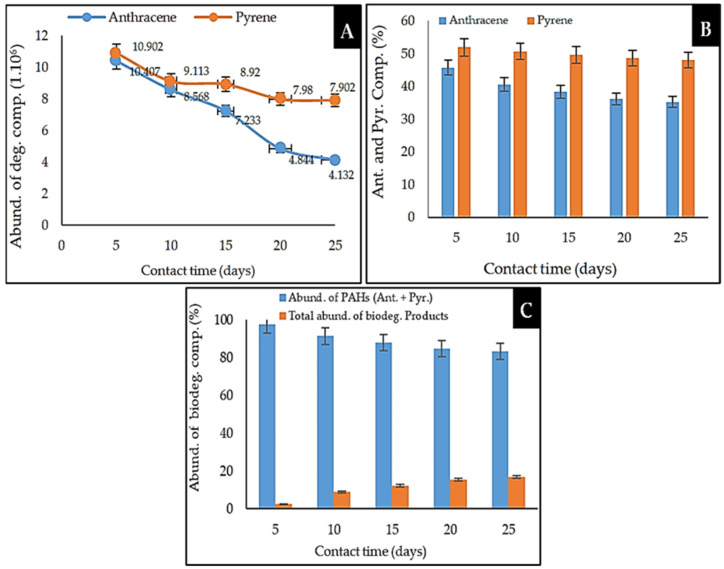
Biodegradation performance of anthracene and pyrene by consortium of marine sponge symbiotic bacteria as a function of contact time. (**A**) Abundance of anthracene and pyrene corresponding to the peak height obtained in the GC–MS analysis; (**B**) reduction in anthracene and pyrene components; (**C**) percentage abundance of anthracene and pyrene.

**Table 1 molecules-26-06851-t001:** Characteristics of seawater at sampling points for sponges around Kodingareng Keke Island.

Sample Code	Coordinate	Salinity (‰)	Temperature ((°C)	pH	TDS (mg/L)	EC (ds/m)	Depth MSL (m)	Distance from the Beach (m)
Sp1	5°6′ 38, 12376″ S	28.3	29.4	7.47	7.41	14.46	3.20	200
	119° 17′70, 76544″ E							
Sp2	5°6′ 11, 62476″ S	28.1	30.9	7.69	7.21	14.20	3.74	250
	119° 17′60, 06228″ E							
Sp3	5°6′ 23, 55372″ S	27.3	30.3	7.70	7.50	12.87	4.25	370
	190° 20′27, 62376″ E							

TDS: total dissolved solid, EC: electrical conductivity, MSL: mean sea level.

**Table 2 molecules-26-06851-t002:** Phenotypic characteristics of marine sponge symbiont bacterial isolates.

Parameters	Marine Sponge Symbiont Bacteria Isolate		
	Sp1 (*Niphates* sp.)	Sp2 (*Hyrtios erectus*)	Sp3 (*Clathria* (*Thalysias*) *reinwardtii*)
Morphology	jagged stem shape, cream color, clustered distribution, endospore is less clear	jagged stem shape, brown color, separate distribution with endospores	round shape, bluish-cream color, clustered distribution, endospores are less clear
Gram staining	reaction (−) with safranin reagent and (−) with 1% KOH alkaline solvent, Gram (+)	fixed color with safranin reagent and (−) with 1% KOH alkaline solvent, Gram (+)	reaction (−) with safranin and (−) with 1% KOH alkaline solvent, Gram (+)
Indole test	−	−	+
Triple sugar iron agar (TSIA)	+	+	−
Nitrate test	+	+	−
Simmons citrate test	−	+	+
Methyl red (Mr) test	−	+	+
Voges–Proskauer (VP) test	+	−	+
Urease test	+	+	+
Provisional guess	*Bacillus* group	*Pseudomonas* group	*Acinetobacter* group

Note: − negative reaction; + positive reaction.

**Table 3 molecules-26-06851-t003:** Genotypic characteristics of marine sponge symbiont isolates by 16S rRNA gene sequence.

Bacterial Isolate	Sequence Samples	Range Sequence Gen-Bank	Identities Quality (%)	Gaps (%)	Species Type
Sp1-Bc	15–967 (952)	710,748–711,700 (952)	922/956 (96.44)	6/956 (0.63)	*Bacillus pumilus* strain GLB197
Sp2-Ps	14–975 (961)	3,666,632–3,667,587 (955)	890/974 (91.14)	30/974 (3.08)	*Pseudomonas stutzeri* strain SLG510A3-8
Sp3-Ac	9–961 (952)	12–964 (952)	951/954 (99.69)	2/954 (0.21)	*Acinetobacter calcoaceticus* strain SLCDA976

**Table 4 molecules-26-06851-t004:** Biodegradation parameters of bacterial consortium suspension toward the PAH contaminants.

Biodegradation Parameters	Interaction Period (Days)					
	0	5	10	15	20	25
Turbidity of interaction media (NTU)	1.01	6.56	12.82	17.62	24.83	26.43
Temperature (°C)	29	29	30	30	30	29
pH	6.67	6.68	6.13	6.10	6.06	6.43
Abundance of gas bubbles	nt	nt	+	++	++	+
Fermentation smell	nt	nt	√	√√	√√	√√

nt: not detected; + gas bubbles appear less abundant; ++ abundance of gas bubbles; √ weak fermentation smell; √√ strong fermentation smell.

**Table 5 molecules-26-06851-t005:** GC–MS data on biodegradation of PAHs by a consortium of marine sponge bacterial symbionts.

Peak Number	Retention Time	Comp. Peak Height (10^6^)	Quality (%)	Total Conc. (%)	Group of Comp.	Approximate Comp. on the Library/ ID
Interaction time 5 days						
1	10.287	0.182	95	1.057	Aromatic	Naphthalene
2	17.468	0.120	47	0.424	Aldehyde	---
3	18.566	10.407	95	45.693	Aromatic **	Anthracene
4	21.819	10.902	96	51.914	Aromatic **	Pyrene
5	24.947	0.223	90	0.912	Alcohol	Phenol
Interaction time 10 days						
1	6.083	0.132	91	0.959	Methyl	Cyclotetrasiloxane
2	17.471	0.794	53	3.752	Aldehyde	---
3	18.559	8.568	95	40.589	Aromatic **	Anthracene
4	21.309	0.239	78	1.401	Aldehyde	---
5	21.809	9.113	96	50.686	Aromatic **	Pyrene
6	24.944	0.185	90	1.070	Alcohol	Phenol
7	26.824	0.240	62	1.543	Carboxylic acid	---
Interaction time 15 days						
1	6.088	0.109	91	0.558	Organosilicon	Cyclotetrasiloxane
2	17.471	0.471	53	1.663	Carboxylate	---
3	18.564	7.233	96	38.311	Aromatic **	Anthracene
4	21.311	0.139	72	0.561	Indole, aldehyde	---
5	21.818	8.920	96	49.619	Aromatic **	Pyrene
6	22.085	0.080	93	0.384	Amide	Hexadecanamide
7	23.637	0.187	91	0.760	Amide	9-octadecetamide
8	23.685	0.123	93	1.084	Amide	9-octadecenamide
9	24.944	0.506	93	1.824	Alcohol aromatic	Phenol
10	26.823	0,700	81	3.236	Carboxylate	---
Interaction time 20 days						
1	16.728	0.131	98	1.858	Alcohol	Benzenemethanol
2	17.467	0.139	53	0.982	Carboxylate	---
3	18.553	4.844	95	36.105	Aromatic **	Anthracene
4	20.329	0.141	46	0.946	Aliphatic	---
5	20.776	0.199	43	1.323	Aliphatic	---
6	21.803	7.980	96	48.603	Aromatic **	Pyrene
7	24.944	0.163	93	1.168	Alcohol aromatic	Phenol
8	26.823	0.219	49	2.016	Carboxylate	Tetraphtelic acid
Interaction time 25 days						
1	17.470	0.242	64	1.112	Methylene	---
2	18.560	4.132	95	35.297	Aromatic **	Anthracene
3	20.332	0.148	47	0.697	Sulfurous acid	---
4	20.778	0.332	47	1.508	Aliphatic	---
5	21.140	0.178	47	1.253	Alcohol aromatic	---
6	21.811	7.902	96	47.959	Aromatic **	Pyrene
7	24.944	0.126	78	0.681	Alcohol aromatic	---
8	26.823	0.234	91	1.504	Carboxylate	Tetraphtelic acid

--- The compound cannot be determined with certainty because the quality was below 90%; ** components of PAHs (anthracene and pyrene) that were degraded.

## Data Availability

Not applicable.

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
