# Peer review of "Anthracene and Pyrene Biodegradation Performance of Marine Sponge Symbiont Bacteria Consortium"

_molecules, 2021, doi:10.3390/molecules26226851_

Round 1

Reviewer 1 Report

The manuscript contains innovative elements. The individual chapters were usually written correctly. The manuscript can be published in Molecules with a few amendments listed below.

  • In Table 1, the abbreviations TDS, DHL and dpl appear for the first time, therefore the abbreviations should be explained below the table. There are no statistical calculations in the table. Please enter homogeneous groups.
  • Line 196 - the beginning of the sentence is wrong.
  • Table 4 - The title of the table is complicated. It is not clear on the basis of which data authors presented the differences in biodegradation identified in the title of the table.
  • Figure 4 - Enter the unit of time on the x-axis and numbers in logarithms or powers on the y-axis. There is no specific unit of measure for abundance. Figures A through E are marked with capital letters, while in the description of the title of the drawing they are marked with lower case letters.
  • Figure 5 - Symbols from a to b in the figure and in the title of the figure should be entered with the same letters.
  • Figure 6 - The description of figures with letters from A to C is presented differently than the description of Figures 4 and 5. Please harmonize.
  • The SI unit should be used instead of ppm in the manuscript.
  • Line 436 - describe the instrument's FTIR parameters.
  • Line 451 - what does the abbreviation cerc.?

Reviewer 2 Report

The manuscript of Marzuli et al aims to explore the potential of a consortium of bacteria isolated from sponges in the degradation of anthracene and pyrene. The scope of this work is of general interest and the data presented is original. However, the language and the organization of the manuscript are very poor, making it difficult to read/follow. There are many incomplete sentences, words repeated in the same sentence and repetition of the some ideas in different sections of the text. It is strongly advisable that authors seek help from an English expert/native speaker. Due to the many inaccuracies found throughout the text, these are not going to be listed here.

General concerns:

-  The lack of information about the methodologies used makes it difficult to evaluate the results/conclusions

- The authors only evaluated the biodegradation of a mixture of anthracene and pyrene. Even though it is true that environmental samples are complex and contain more than one compound, the biodegradation of each PAH should have been assessed individually to provide information on how the presence of other compounds affects the degradation of each compound.

Other concerns:

Introduction – The main goal of the work is not clearly stated in the end of the introduction.

Line 79 – what do you mean by “mucus substances that act as enzymes”? Do you mean that there are enzymes trapped in the mucus substance? Please clarify

Line 110/111 – What do you mean? Please clarify this sentence

Line 131 – replace “..the potential symbiosis of ….hydrocarbon components” with “the potential of the symbiotic bacteria in the biodegradation of hydrocarbon components”.

Table 1 – Add a footnote defining the abbreviations TDS, DHL, dpl

Figures 1, 2 and 3 – These figures should be combined into one. They provide similar information for three different sponges.

Section 2.3 – Describe the phenotypic characteristics used to identify the bacteria and provide the data (e.g. figure)

Line 184 – The urea test does not indicate carbon, it detects the alkaline fermentation of urea with the resultant production of ammonia.

Lines 184 – not all bacteria tested showed a positive result in the citrate test. Please rephrase this sentence

Lines 187-189 – What kind of phenotypic traits lead to the identification of the bacteria? Please clarify this

Section 2.4 – How were bacteria identified? Please add that information.

Section 2.5.3 – This section is redundant and should be included in the previous one. Figure 4 shows the same data as table 5. Please remove it.

Line 301 – replace “An-trasena and Pyrena” with “anthracene and pyrene”

Figure 6 – Fig6 A is redundant. That information is already present in table 5.

Line 325 – What do you mean with “showed good quality”? What are the critearia used for this assessment?

Lines 332-337 – This sentence is incorrect: a phenotypic characterisation of a bacteria consortium does not provide information/suggests a mechanism of biodegradation.

Discussion – add a final paragraph with the conclusions of the work

Material & Methods – This section is very incomplete. For example, there is no information about how genotypic analysis was made, including how was DNA extracted, what primers were used, how were sequences obtained, etc…

Round 2

Reviewer 2 Report

I would like to thank the authors for the responses to the comments/questions. The authors have addressed most of the questions. However, in my opinion, the manuscript can be further improved.

General comment: As stated before, the English throughout the manuscript is poor, making it difficult to read/Follow. It is strongly advisable that authors seek help from an English expert/native speaker.

Line 124 – remove “without analysing….degradation product”

Figures 1-3 – Please remove description of results from the figures legend. The figures can be separated, but can also be combined into one figure composed of part A, B, and C.

Section 2.3 – The phenotypic characterization performed by the authors is based on two separate aspects: morphological traits and biochemical tests. The fact that the authors use the expressions “phenotypic test” or “phenotypic results” to infer the capacity of the strains to degrade PAHs, is confusing, since no such information can be retrieved from the morphological analysis. Please, clarify the terms used along the text.

Line 217 – It is incorrect to state that a positive result in citrate test indicates that bacteria can “make carbon”. This test indicates that the bacteria can convert citrate to pyruvate, which enters the organism’s metabolic cycle for the production of energy. Please correct this.

Line 230 – Please replace “urea test” by “urease test”.

Lines 252-253 – The information added to this section is not pertinent. The important thing here is to indicate the molecular analysis was done using the 16S rRNA sequence.

Figure 4 – the authors stated that Figure 4 was removed. However, the figure is still included in the manuscript.

Figure 5 – Please rename it Figure 4.

Figure 6 - Please rename it Figure 5. Panel A: please add in the legend that the values for the abundance of anthracene and pyrene correspond to the peaks high obtained in GC-MS analysis. Panel C: the Y axis is defined as “Degr. Comp (%)”. However, for one of the data sets, namely “Total PAHs (Ant + Pyr)”, the values correspond to % of abundance and not % of degradation. Please correct that.

Line 401 – The authors state that the “decrease in the abundance of anthracene components reached 60,30%, while pyrene was only 27,52%”. How were the values obtained? They do not match those in graphic of panel B.

Line 430 – Please replace phenotypic with biochemical.

Author Response

Dear Reviewer 2

We have revised our manuscript based on your comment in Round 2. Please find our PDF file attached to this mail. We have submitted this manuscript for language editing by the language editing service that the MDPI Molecules journal provides. Would you please give us a way to submit our revised for Round 2? Hopefully, our manuscript can be considered acceptable to published. Thank You.

Sincerely yours,

Ismail Marzuki
